# A Possible Modulator of Vitiligo Metabolic Impairment: Rethinking a PPARγ Agonist

**DOI:** 10.3390/cells11223583

**Published:** 2022-11-12

**Authors:** Federica Papaccio, Barbara Bellei, Monica Ottaviani, Andrea D’Arino, Mauro Truglio, Silvia Caputo, Giovanni Cigliana, Lorenzo Sciuto, Emilia Migliano, Alessia Pacifico, Paolo Iacovelli, Mauro Picardo

**Affiliations:** 1Cutaneous Physiopathology and Integrated Center of Metabolomics Research, San Gallicano Dermatological Institute, IRCCS, 00144 Rome, Italy; 2Clinical Pathology Unit, Department of Research, Advanced Diagnostics, and Technological Innovation, Translational Research Area, IRCCS Regina Elena National Cancer Institute, 00144 Rome, Italy; 3Plastic and Regenerative Surgery, San Gallicano Dermatological Institute, IRCCS, 00144 Rome, Italy; 4Phototherapy Unit, San Gallicano Dermatological Institute, IRCCS, 00144 Rome, Italy

**Keywords:** vitiligo, PPARγ, pioglitazone, melanocytes, fibroblasts, cellular metabolism, glucose metabolism, metabolic impairment

## Abstract

Vitiligo is a complex disease wherein derangements in multiple pathways determine the loss of functional melanocytes. Since its pathogenesis is not yet completely understood, vitiligo lacks a definitive safe and efficacious treatment. At present, different therapies are available; however, each modality has its baggage of disadvantages and side effects. Recently we have described several metabolic abnormalities in cells from pigmented skin of vitiligo patients, including alterations of glucose metabolism. Therefore, we conducted a study to evaluate the effect of Pioglitazone (PGZ), a Peroxisome proliferator-activated receptor-γ (PPARγ) agonist, on cells from pigmented vitiligo skin. We treated vitiligo melanocytes and fibroblasts with low doses of PGZ and evaluated the effects on mitochondrial alterations, previously reported by our and other groups. Treatment with PGZ significantly increased mRNA and protein levels of several anaerobic glycolytic enzymes, without increasing glucose consumption. The PGZ administration fully restored the metabolic network, replacing mitochondrial membrane potential and mitochondrial DNA (mtDNA) copy number. These effects, together with a significant increase in ATP content and a decrease in reactive oxygen species (ROS) production, provide strong evidence of an overall improvement of mitochondria bioenergetics in vitiligo cells. Moreover, the expression of HMGB1, Hsp70, defined as a part of DAMPs, and PD-L1 were significantly reduced. In addition, PGZ likely reverts premature senescence phenotype. In summary, the results outline a novel mode of action of Pioglitazone, which may turn out to be relevant to the development of effective new vitiligo therapeutic strategies.

## 1. Introduction

Vitiligo is an acquired chronic pigmentation disorder characterized by the appearance of white spots on the skin due to melanocyte loss. Affecting 0.5–1% of the population worldwide, it is the most diffused depigmentation disorder [1,2]. 

The etiology of the disease is multifactorial and presents different manifestations, progression rates, and responses to treatment [3]. Multiple mechanisms are supposed to be involved in melanocyte disappearance. These include genetic predisposition, autoimmune responses, oxidative stress, environmental triggers, metabolic abnormalities, impaired renewal of melanocytes, and altered inflammatory state [4]. However, the overall contribution of each of these processes is still under debate, suggesting that multiple mechanisms may work jointly in vitiligo to contribute to the destruction of melanocytes. 

We previously demonstrated that melanocytes, obtained from pigmented vitiligo skin, show cellular and molecular alterations. In particular, we described a constitutive activation of antioxidant enzyme genes and defects in mitochondrial metabolism, reflected in an altered expression and activity of complex I, increased generation of reactive oxygen species, low ATP production, and modified expression of some glycolytic enzymes [5,6]. The imbalance in the pro and anti-oxidative state is responsible for the increased susceptibility to external stressful stimuli and for the premature senescence of the skin characterized by the production of different proteins of the so-called Senescence-Associated Secretory Phenotype (SASP) [5] and increased cholesterol content in vitiligo melanocytes cultures [5]. 

Recently, our group has shown that some of the described alterations are not exclusive to melanocytes but can be observed in other skin cells obtained from pigmented areas, suggesting that vitiligo leads to the degeneration of the entire skin [7,8]. Fibroblasts exhibit oxidative stress, overexpression of p53, and a senescent phenotype [7]. These features are the basis for the altered secretion of soluble growth factors supporting melanocyte survival and homeostasis. The consequent alteration of the dermal-epidermal network could be the basis for melanocyte detachment [7,8,9]. Moreover, our group recently demonstrated abnormalities in the keratinocyte differentiation process and consequently an inappropriate assembly of the epidermal layers which, following stressful events, can induce an inflammatory reaction capable of activating an immune response targeting melanocytes [8]. These findings further strengthen the pivotal role of metabolic alterations in vitiligo onset and development.

Lastly, our vitiligo unit carried out a retrospective analysis underlining that vitiligo patients frequently display modest alterations of the spectrum of the metabolic syndrome [10]. Specifically, while in vitiligo patients, lower levels of total cholesterol were measured overall, the distribution of lipid subpopulations was less favorable with respect to the healthy controls. In particular, triglycerides and LDL cholesterol showed greater values while HDL cholesterol levels were consistently lower than healthy controls. Likewise, fasting blood glucose levels were significantly higher in vitiligo patients, even if below the diabetes threshold and within the range of impaired fasting glucose [10]. PPARγ is a ligand-activated transcription factor, belonging to the PPARs nuclear receptor superfamily, and is considered one of the master modulators of mitochondrial biogenesis and function [11]. After the heterodimerization with the retinoid X receptors (RXRs), PPARγ produces functional transcription factors contributing to the trans-activation of key genes involved in energy homeostasis and cellular differentiation [12]. A crucial role for PPARγ in fat cell differentiation, lipid storage, vascular function, and energy metabolism has been identified [13]. Thiazolidinediones (TZDs) represent a class of compounds that are high-affinity ligands for the transcription factor PPARγ [14]. Among the TZD, PGZ is commercially available and selectively stimulates PPARγ [15]. PGZ is best known for its use in improving glycemic control in adults with type 2 diabetes mellitus and reducing insulin resistance. Moreover, it can normalize hyperglycemia induced by intracellular ROS and mitochondrial ROS (mtROS) production [16]. PGZ is effective in many important biologic processes, including inflammation [17] and beneficial effects have been described in the treatment of psoriatic patients, decreasing the production of inflammatory mediators, such as Interleukin 6 (IL-6) [18]. PGZ can act as an immune-modulating agent inhibiting the proliferation and differentiation of keratinocytes in psoriasis [18]. According to the idea that vitiligo cells carry metabolic alterations, PGZ was tested on melanocytes and fibroblasts from pigmented skin of vitiligo subjects, in order to evaluate whether PPARγ activator is able to rescue metabolic impairment in vitiligo and consequently improve the biological behavior of melanocytes. Our study also provides proof-of-concept evidence that PGZ contributes to counteracting the premature aging phenotype, characteristic of vitiligo skin. This study aims to clarify that increased awareness of the metabolic aspects of vitiligo could be considered crucial to advance novel and alternative therapeutic target options in vitiligo treatment.

## 2. Materials and Methods

### 2.1. Ethic Statement

The Declaration of Helsinki Principles was followed, and patients were given written informed consent. The Institute’s Research Ethics Committee (Regina Elena Institute and San Gallicano Dermatological Institute) approval was obtained to collect samples of human material for research (Prot CE/751/16 approved on 12 January 2016).

### 2.2. Skin Biopsies and Cell Cultures

The cell lines obtained from 10 vitiligo and 10 normal subjects, age and sex-matched, were used in the study. The control samples, normal human primary epidermal melanocytes (NHM), were obtained from subjects who underwent plastic surgery for diseases unrelated to pigmentation disorders. The primary epidermal melanocytes (VHM) and fibroblasts (VHF) from vitiligo subjects were isolated from 1 cm^2^ skin biopsy in a non-lesional area. Briefly, the skin was cut into approximately 4 mm^2^ sized pieces and incubated overnight at 4 °C with dispase (2.5 mg/mL) to separate the epidermis from the dermis. The dermis was digested with collagenase 0.35% for 1 h at 37 °C. Isolated NHM and VHM were cultured in 254 Medium (Cascade Biologics, Portland, OR, USA; ThermoFisher, Waltham, MA, USA) supplemented with a specific Growth Factors cocktail (Cascade Biologics) and penicillin/streptomycin (Gibco Hyclone Laboratories, South Logan, UT, USA). Isolated VHF were cultured in DMEM (EuroClone S.p.A., Milan, Italy) supplemented with 10% fetal bovine serum (FBS) (EuroClone S.p.A., Milan, Italy) and antibiotics (Hyclone Laboratories, South Logan, UT, USA). All the aforementioned analyses were performed between 2 and 7 culture passages.

### 2.3. Pioglitazone (PGZ) Treatment

PGZ (Merck, Sigma-Aldrich, Merck KGaA, Darmstadt, Germany) was dissolved in dimethyl sulfoxide (DMSO) (Merck, Sigma-Aldrich, Merck KGaA, Darmstadt, Germany) to a stock solution of 20 mM and added to the cell growth medium at the final concentrations of 2µM for periods from 6 h to 10 days, according to the type of parameter evaluated. In untreated control cells, an equal volume of DMSO was added.

### 2.4. Proliferation Assay

In this case, 10^5^ of vitiligo melanocytes were seeded in 6-well plates and treated with 2 μM of PGZ or DMSO. After 48 h and 10 days, cells were washed with PBS, trypsinized, centrifuged at 800 rpm, resuspended and then counted by TC20TM Automated Cell Counter (BIO-RAD). 

### 2.5. Semi-Quantitative Real-Time Polymerase Chain Reaction (RT-PCR)

The total RNA was extracted from each cell line using Aurum Total mini kit (Bio-Rad Laboratories, Milan, Italy). Here, cDNA was synthesized from 1 μg of total RNA using the PrimeScriptTM RT Master Mix (Takara Bio Inc., Beijing, China) according to the manufacturer’s instructions. Quantitative real-time RT-PCR was performed in a reaction mixture containing SYBR qPCR Master Mix (Vazyme Biotech Co., Ltd., Nanjing, China) and 25 pmol of forward and reverse primers. The reactions were carried out using a CFX96 Real-Time System (Bio-Rad Laboratories). All samples were run in triplicate. The amplification of the β-Actin (β-act) from each sample has been used as the internal control. For each gene, the assessment of quality was performed by examining PCR melt curves after quantitative (q)RT-PCR to ensure product specificity. Appendix A shows the oligonucleotide sequences used to detect the expression of reported target genes. 

### 2.6. Western Blot Analysis

The cell extracts were prepared with RIPA buffer containing proteases and phosphatase inhibitors. The proteins were separated on SDS-polyacrylamide gels, transferred to nitrocellulose membranes and then treated with the primary antibodies reported in Appendix A. Horseradish peroxide-conjugated goat anti-mouse or goat anti-rabbit secondary antibody complexes were detected by chemiluminescence (Cell Signaling Technology, Danvers, MA, USA). Imaging and densitometry analyses were performed with the UVITEC Mini HD9 acquisition system (Alliance UVItec Ltd., Cambridge, UK).

### 2.7. ATP Determination

The intracellular level of ATP was measured using a commercial fluorimetric kit (ThermoFisher Scientific) according to the manufacturer’s instructions. The results were normalized for the number of cells contained in each sample and reported as μM. The measurement was performed in duplicate for each sample and the experiments were repeated twice.

### 2.8. Glucose Determination 

The extracellular level of glucose was measured using Roche/Hitachi cobas c 503 according to the manufacturer’s instructions. The results were reported as mg/dL mean value.

### 2.9. mtDNA Quantification

The total DNA was prepared from melanocytes using DNeasy Blood and Tissue (Qiagen GmbH, Hiden, Germany) according to the manufacturer’s recommendations and stored at −20 °C. mtDNA content was measured by real-time PCR using a CFX96 Real-Time System (Bio-Rad Laboratories). The amplification conditions were as follows: 5 min at 95 °C, then 45 cycles of 15 s at 95 °C and 1 min at 58 °C. A dissociation curve was also calculated for each sample to ensure the presence of a single PCR product. The experiment was performed in triplicate. The relative quantification of mitochondrial DNA (mtDNA) over nuclear DNA (nuDNA) levels was determined using the difference in the threshold cycle values of the nuclear TATA-box-binding protein region on chromosome 6 and the mitochondrial non-coding control region D-loop (ΔCt, namely, CtmtDNA−CtnuDNA). The relative abundance of the mitochondrial genome was reported as 2−ΔCt. The primers used were the following: 

mtDNA forward, GATTTGGGTACCACCCAAGTATTG (SEQ ID NO:15); 

reverse, GTACAATATTCATGGTGGCTGGCA (SEQ ID NO:16); 

nuDNA forward, TTCCACCCAAGTATTG (SEQ ID NO:17); 

reverse, TGTTCCATGCAGGGGAAAACAAGC (SEQ ID NO:18) 

### 2.10. Protein Determination by Sandwich Enzyme-Linked Immunosorbent Assay (ELISA)

IL-6 determination in the supernatants of treated and untreated vitiligo melanocytes was quantified by ELISA assay (Aviva System Biology, San Diego, CA, USA) according to the manufacturer’s protocol. The supernatants were collected after 48 h of treatment. The results were normalized for the number of cells contained in each sample and were expressed as picograms per milliliter (pg/mL). The measurement was performed in duplicate for each sample and the experiments were repeated twice. 

### 2.11. Detection of Intracellular ROS Levels

The production of ROS has been assessed with the fluorescent dye 2′7′-dichlorodihydrofluorescein diacetate (H_2_DCFDA; Sigma-Aldrich). Cell permeable, non-fluorescent H_2_DCF is oxidized to highly fluorescent dye 2′7′-dichlorofluorescein (DCF) in the presence of intracellular ROS. The cells were incubated with 2.5 μmol L−1 H_2_DCF for 30 min at 37 °C and 5% CO_2_ in phenol red-free full-starved medium in the dark. After removing the probe solution, cells were washed with PBS, trypsinized, centrifuged at 800 rpm, and then resuspended in PBS. After the oxidation of H_2_DCF into fluorescent DCF by ROS, signals were measured by MACSQuant Analyzer 10 Flow Cytometer (Miltenyi Biotec, Bergisch Gladbach, Germany). The data were collected from three independent experiments.

### 2.12. Assessment of Mitochondrial Membrane Potential (∆Ψ) 

The cells were incubated with 2 μM of dye JC-1 (5′,6,6′-tetrachloro-1,1′,3,3′-tetraethylbenzimidazolylcarbocyanine iodide) (ThermoFisher Scientific) for 30 min at 37 °C and 5% CO_2_ in phenol red-free full-starved medium in the dark. After removing the probe solution, the cells were washed with PBS, trypsinized, centrifuged at 800 rpm, and then resuspended in PBS. The double fluorescence staining of mitochondria by JC-1, either as green fluorescent J-monomers or as red fluorescent J-aggregates, was used for monitoring the mitochondrial membrane potential. The signals were measured by MACSQuant Analyzer 10 Flow Cytometer (Miltenyi Biotec). The data were collected from three independent experiments.

### 2.13. Mitochondrial Mass Measurement 

The cells were incubated with 0.1 μM of MitoTracker^®^ probe (ThermoFisher Scientific) for 30 min at 37 °C and 5% CO_2_ in phenol red-free full-starved medium in the dark. After removing the probe solution, cells were washed with PBS, trypsinized, centrifuged at 800 rpm, and then resuspended in PBS. Fluorescence signals were measured by MACSQuant Analyzer 10 Flow Cytometer (Miltenyi Biotec). The data were collected from three independent experiments.

### 2.14. PCA Analysis

The Principal Component Analysis (PCA) was performed on the gene expression profiles, using Singular Value Decomposition (SVD) of the data to project it to a lower dimensional space. The input data were centered but not scaled for each feature before applying the SVD, using the Linear Algebra PACKage (LAPACK) implementation in the Python library scikit-learn. PCA analysis was performed on the genes reported in Appendix A.

### 2.15. Statistical Analysis

The results in the figures are representative of several experiments performed with at least six cell lines from independent donors. Student t-test was used to assess the statistical significance with thresholds of * *p* ≤ 0.05 and ** *p* ≤ 0.01. 

## 3. Results

### 3.1. Effects of Pioglitazone on Melanocytes Glucose Metabolism

Since prior observations reported that glucose uptake is greatly increased in vitiligo melanocytes [6], we measured the glucose amount in the medium of cultured vitiligo melanocytes both treated with or without PGZ. As shown, the glucose medium content was increased by treatment; consequently, PGZ-treated vitiligo melanocytes presented a lower glucose consumption compared to the untreated ones after 48 h (Figure 1A).

Consistent with this decrease, PGZ diminished the gene expression levels of glucose transporters Glut1 and Glut4 after 6 h, while did not seem to modify the expression of Glut3 (Figure 1B). We next investigated whether the expression of enzymes involved in glucose metabolism was adjusted by PGZ. Using the real-time evaluation, we analyzed the gene expression levels of some glycolytic enzymes involved in anaerobic respiration after 6 h of PGZ treatment. In sharp contrast, all the different enzymes evaluated were induced by treatment. Hexokinase2 (Hexo2), Pyruvate kinase isozymes M1/M2 (Pkm1,2), and subunit 4 of Pyruvate dehydrogenase complex (Pdk4) resulted significantly upregulated by PGZ (Figure 1C). Furthermore, Western blot analysis confirmed the induction of glycolytic enzymes by PGZ treatment (Figure 1D). To further corroborate our analysis, we conducted a measure of β-oxidation key enzymes, such as Acyl-CoA Dehydrogenase medium-chain (ACADM), Acyl-CoA Dehydrogenase (ACADS), Peroxisomal Acyl-CoA Oxidase (ACOX). A slight significant threshold decrease in the gene expression levels of this class of enzymes was observed (Figure 1E). Similarly, we measured a substantial reduction of enzymes involved in the Krebs cycle, as well as Pyruvate Carboxylase (PC) and Alpha-ketoglutarate (OGDH) (Figure 1F), confirming PGZ’s selective activity on pathways implicated in ATP production. It is well documented that the mammalian target of rapamycin (mTOR) and AMP-activated protein kinase (AMPK) signaling pathways play a central role in response to ATP levels [19]. Recently, our group demonstrated that, in vitiligo melanocytes, the expression of AMPK phosphorylated (pAMPK) was upregulated, whereas the phosphorylation level of the downstream target of mTORC1, S6 kinase (pS6K), was reduced [20], further supporting the impaired metabolic condition. To better affirm the effect of Pioglitazone on cellular metabolism, we evaluated the effect of treatment on these pathways. Thus, 48 h of treatment with PGZ induced dephosphorylation of AMPK leading to higher phosphorylation of mTOR and S6 (Figure 1G). Taken together, these findings highlight that both acute and prolonged Pioglitazone administration improves glucose metabolism without an effectively increased consumption. Moreover, PGZ seems to be able to revert signaling pathways that regulate the intracellular metabolic state.

### 3.2. Pioglitazone Attenuates Altered Melanocytes Mitochondrial Condition

Starting from previously and currently reported mitochondrial alterations, we examined the role of PGZ in the recovery of mitochondrial structural and functional parameters. RT-PCR analysis showed a significant increase in the mtDNA copies in treated vitiligo melanocytes. The relative increment of mtDNA content was normalized on the nuDNA (Figure 2A).

Similarly, we observed that the gene expression of PGC1α, a key factor in mitochondrial biogenesis, was induced by treatment (Figure 2B). As previously reported, vitiligo melanocytes are characterized by a higher mitochondrial mass, as a compensatory mechanism to obtain a sufficient energetic level in a steady-state condition [6]. Here, we did not measure a mitochondrial mass decrease after 10 days of treatment (Figure 2C). Instead, q-RT-PCR underlined a reduction of voltage-dependent anion-selective channel 1 (VDAC1) (Figure 2D). Additionally, we did not find increased protein levels of total mitochondrial OXPHOS complexes (complex I, complex II, complex III, complex IV, and complex V) (Figure 2E). Our findings highlighted the ability of PGZ to modulate the processes that vitiligo cells act to compensate for mitochondrial defects. To further strengthen the effective ability of PGZ to reverse the altered phenotype of vitiligo melanocytes, we performed a principal component analysis (PCA). PCA was generated by comparing 6 primary treated vitiligo melanocytes with the same number of primary untreated vitiligo and primary untreated normal melanocytes. The PCA plot was based on the fold change of all the genes surveyed in the present study. As shown in Figure 2F, the melanocytes from normal individuals clustered closely, whereas the melanocytes from vitiligo patients are spread out, representing the heterogeneous signatures of these cells [21]. Moreover, a clear similarity is evident between treated vitiligo melanocytes and the normal controls. Collectively, our evidence supports the idea that aberrant mitochondrial components could be restored by Pioglitazone. 

### 3.3. Pioglitazone Improves the Whole Energetic Status in Vitiligo Melanocytes

The cumulative evidence from our and other research groups demonstrated that melanocytes from pigmented skin of vitiligo patients present some structural and functional alterations and are characterized by an impaired energetic metabolism [5,6,22]. The bioenergetic deficit involves both oxidative phosphorylation and ATP synthesis due to a defect in the mitochondrial complex activity [6]. Once verified that 2 µM of PGZ was the lowest concentration able to induce a significant increase in cell proliferation rate after 48 h and 10 days of treatment (Figure 3A), this concentration was used in all experiments.

Firstly, we evaluated the effect of the treatment on mitochondrial bioenergetic status. We measured ATP content in treated and untreated cells. An increase in ATP levels was detected in melanocytes treated for 10 days (Figure 3B). Maintaining the mitochondrial membrane potential (ΔΨm) is required for ATP production. To evaluate if PGZ is able to restore mitochondrial membrane potential, which is altered in vitiligo melanocytes, we compared treated vitiligo melanocytes with untreated controls (Figure 3C). Following 10 days of incubation with PGZ, flow cytometry analysis revealed an increase in mitochondrial membrane potential expressing as a ratio green/red. Since lower ΔΨ and decreased activity of the respiratory chain is observed with a simultaneous increase in reactive oxygen species production, we investigated the effects of PGZ on intracellular ROS in vitiligo cells using the redox-sensitive probe DCFH-DA. DCFH-associated fluorescence was decreased in vitiligo melanocytes after treatment with PGZ for 10 days (Figure 3D). Parallelly, the mRNA of antioxidant enzymes such as Catalase (Cat), heme oxygenase (HO-1), and superoxide dismutase 2 (SOD-2) were increased by PGZ, even without being statically significant, possibly due to the variability of the basal level of expression (Figure 3E). The upregulation of mitochondrial uncoupling proteins is another potential explanation for the reduced mitochondrial efficiency [23]. Thus, we evaluated the effect of PGZ on Uncoupling protein 2 (UCP2), a member of the uncoupling mitochondrial transmembrane proteins family. RT-PCR analysis revealed that Pioglitazone is capable to reduce UCP2 gene expression already after 6 h (Figure 3F). This modulation was then confirmed at the protein level (Figure 3F). These findings suggest that the impaired functionality of mitochondria in vitiligo could be ameliorated by a low dose of Pioglitazone. 

### 3.4. Pioglitazone Reduces MITF and Senescence-Associated Markers Expression in Vitiligo Melanocytes

An additional intrinsic abnormality in vitiligo melanocytes, compared to the normal counterpart, is the marked higher expression of the Microphthalmia-associated transcription factor (MITF), which is associated with a differentiated status (Figure 4A). 

Here, we showed that PGZ significantly decreased gene expression of MITF after 6 h (Figure 4B). However, we did not detect a similar reduction at the protein level, after 48h (Figure 4C). Many reports have suggested that the ERK signaling pathway has a critical role in the regulation of melanogenesis and it has been reported that ERK activation leads to MITF phosphorylation and its subsequent degradation [24,25,26]. To better describe the effect of PGZ on MITF expression, we examined ERK phosphorylation levels up to 48 h of treatment. As shown in Figure 4D, PGZ induced de-phosphorylation of ERK and consequently MITF activation, contrary to the gene expression level. Kim et al. reported that inhibition of ROS and ERK abolished the degradation of MITF [27], similarly the inhibitory effect on ROS production by PGZ treatment could lead to MITF upregulation in our model, explaining its dual action. In vitiligo melanocytes, the premature senescence phenotype is characterized by the production of different proteins of the so-called senescence-associated Secretory Phenotype (SASP), including cytokines, growth factors, and molecules implicated in cell adhesion and tissue remodeling [5]. Next, we tested whether PGZ treatment was capable to alter the premature senescence phenotype in vitiligo melanocytes. Our results indicated that PGZ significantly reduced insulin growth factor binding proteins 3 (IGFBP3), both at mRNA and protein levels (Figure 4E). Additionally, insulin growth factor binding proteins 7 (IGFBP7) and p16 gene expression levels were diminished by PGZ (Figure 4E). Furthermore, the ELISA assay verified that PGZ also inhibited interleukin 6 levels (IL-6), a senescence-associated inflammatory mediator (Figure 4F). Thus, Cyclooxygenase-2 (Cox-2), an inducible enzyme, involved in stress-induced senescence and upregulated in vitiligo melanocytes [5], was feebly, but significantly, downregulated by PGZ (Figure 4G). 

### 3.5. Pioglitazone Modulates the Pro-Inflammatory Mediators in Vitiligo Melanocytes

Vitiligo pathogenesis involves complex combinatorial factors, including dysregulation of both innate and adaptive immune responses [2,28]. Several groups have demonstrated that inhibition of the immune response could appear as a promising strategy for vitiligo treatment [28]. Increased oxidative stress can induce the production of pro-inflammatory cytokines and activation of signals important for the induction of the immune system in vitiligo. Regarding the adaptive immune response, we have detected that PGZ reduced the gene expression of CXCR3B-isoform of the chemokine receptor (Figure 5A), whose expression is upregulated in vitiligo melanocytes [29].

Notably, PGZ also inhibited the release of HMGB1, which is capable of activating the immune response (Figure 5B). In vitiligo, melanocytes increase the expression of Heat shock protein 70 (Hsp70) [30,31], a stress-inducible protein belonging to the heat shock protein family. Here we detected a downregulation of Hsp70 mediated by PGZ at the protein level (Figure 5C). Moreover, PGZ significantly lowered gene expression levels of stress molecule MHC class 1 chain-related protein A and B (MICA/MICB) (Figure 5D), NKG2D ligands, whose role was recently described in vitiligo pathogenesis [32]. The PD-1/PD-L1 pathway is classically associated with the modulation of the immune response of T cells, and the binding of PD-1 to a ligand can inhibit the T-cells proliferation and secretion of cytokines [33]. Interestingly, we measured, in vitiligo melanocytes compared to the normal controls, an overexpression of PD-L1 (Figure 5E), which can be expressed by melanocytes, especially in inflammatory environments [34]. As well, PGZ treatment seems to significantly reduce PD-L1 mRNA and protein levels (Figure 5F). The data collected propose a novel possible PGZ target, highlighting the synergistic interplay between metabolic alteration and inflammatory process in vitiligo pathogenesis.

### 3.6. Pioglitazone Ameliorates Intrinsic Alterations of Vitiligo Fibroblasts

Our earlier studies revealed functional and metabolic alteration in dermal fibroblasts, as well as myofibroblast and premature senescence phenotype [5,7]. To further characterize PGZ action in vitiligo cells, we performed a similar analysis on vitiligo fibroblasts. In treated vitiligo fibroblasts, we confirmed an increase in anaerobic glycolysis processes (Figure 6A) without an increment of glucose consumption, as proved by the glucose increased amount in fibroblasts treated medium (Figure 6B).

Furthermore, we detected a significant reduction of IGFBP3 gene expression and protein levels after PGZ treatment (Figure 6C). Similarly, mRNA levels of IGFBP5, p16 and p21 appeared downregulated by PGZ administration (Figure 6D). Most recently, it was reported that perturbation of glycolysis and fatty acid oxidation (FAO) pathway enzymes reveals their reciprocal effects in extracellular matrix components (ECM) upregulation and downregulation [35]. Instead, an enhanced release of growth factors and messengers that are part of a paracrine signaling network that controls melanocyte function is ascribed to vitiligo fibroblasts [7]. Starting from these two different pieces of evidence, to better elucidate a potential effect of PGZ in vitiligo cells, we focused our analysis on peculiar vitiligo fibroblast characteristics. We verified that PGZ treatment decreased the expression of several growth factors as well as Hepatocyte growth factor (HGF), both mRNA and protein levels (Figure 6E), and gene expression of Stem cell factor (SCF) (Figure 6E) and Vascular endothelial growth factor (VEGF) (Figure 6E). Furthermore, the Western blot analysis also revealed a decreased protein level of basic fibroblast growth factor (b- FGF) (Figure 6E). Conversely, the gene expression and protein levels of Fibronectin resulted increased by treatment (Figure 6F). Here, we proved that the modulation of metabolism could modify cellular phenotype and that PGZ could revert the altered expression of growth factors, involved in melanocyte homeostasis. The expression of transmembrane glycoprotein CD36, which imports long-chain fatty acids intracellularly for FAO, has been demonstrated to be inversely correlated with ECM abundance in normal skin and be downregulated in skin fibrosis [36]. Therefore, we evaluated whether CD36 could be modified by PGZ in vitiligo cells. In contrast with this, the CD36 gene and protein expressions were upregulated by PGZ in vitiligo fibroblasts and melanocytes (Figure 6G). These data show that PGZ action on the anaerobic glycolytic pathway could adjust aberrant vitiligo fibroblasts phenotype, also modulating the production of paracrine signals which contribute to melanocytes functionality.

## 4. Discussion 

Our data clearly demonstrate that, in vitro, the Peroxisome proliferator-activated receptors-γ agonist, Pioglitazone, significantly embanks the underlying cellular damage in non-depigmented skin of vitiligo patients. Compelling evidence, both at a cellular and molecular level, indicates that cumulative defects cause sickness in the skin [37]. Among the various pathogenetic hypotheses, our research has been particularly effective in the explanation of non-immunological factors, demonstrating the presence of metabolic impairment in almost all epidermal cells [5,7]. In accordance with our data, vitiligo could be defined as a mitochondrial disease leading to an aging phenomenon and to an alteration of the skin barrier, which could be an initial event in the development of the inflammatory process [5,6,8,38]. The regulation of energy metabolism is critical in maintaining intracellular redox equilibrium, strictly related to mitochondria efficiency. PPARs are nuclear receptors, which have an essential role in the mammalian physiological system [39]. PPARγ, the most widely investigated subtype, is involved in regulating glucose and lipid metabolism. In the skin, it manages the expression of a class of genes involved in cell proliferation, differentiation, and inflammatory responses [40]. Studies investigating its regulatory role on inflammatory markers such as IL-6, IL-17 and TNF-α, suggest that PGZ may be an effective candidate for autoimmune diseases and inflammatory skin disorders [41]. While the anti-diabetic effects of PPARγ agonists thiazolidinediones are well established, the full spectrum of action is not fully understood, especially when the effects on mitochondrial function are considered. To that end, we evaluated the drug’s effect on the metabolic features of vitiligo cells, trying to highlight whether an improvement of the cellular metabolic status was associated with a significant modification of the biological behavior. All data presented, have precisely clarified how the known activity of PGZ is capable of shifting the metabolism of vitiligo cells towards the aerobic end of the spectrum with a consequent appropriate use of glucose, reverting the aberrant metabolic abnormalities observed. Specifically, we verified that acute PGZ administration reduced glucose consumption, meanwhile clearly increasing the expression of several anaerobic glycolytic enzymes. Likewise, the prolonged treatment decreased reactive oxygen species content and restored the physiological membrane potential. The acute metabolic effects of PGZ also reduced the phosphorylation of AMPK, a key cellular energetic sensor [42]. When cellular energy is low, as well as in vitiligo cells, AMPK is activated and targets a range of physiological processes, which finally concur in increased energy production and a coordinated decrease in ATP usage [43]. Moreover, it is known that hyperactive AMPK has been linked to age-related diseases, such as Alzheimer’s disease and other cognitive dysfunctions [44]. Therefore, the inactivation of AMPK signaling, induced by PGZ treatment, confirmed the possibility of reversing the senescent changes we described in previous works. As mentioned above, Dell’Anna et al., stated a mitochondria failure associated with the peroxidation of the mitochondrial membrane due to the ROS generation and the overexpression of glycolytic enzymes to try to compensate for the decreased ATP generation via mitochondrial respiration. The disclosed increase of PGC1α and mitochondrial DNA copies suggest a mitochondria genesis induced by PGZ with the rescue of the ATP generation. In addition, our study confirmed previous reports [45,46], proving the anti-oxidative capability of PGZ. Collectively, our experimental evidence showed, that by raising metabolic spectrum, diminishing oxidative stress, and improving ATP production, PGZ seems to render vitiligo cells more similar to normal controls, as also confirmed by PCA analysis performed. MITF is a chief transcriptional regulator of melanogenesis and is crucial for developing melanocytes [47]. In vitiligo melanocytes, we observed MITF gene overexpression, indicating the activation of the differentiation processes. However, the protein level of MITF increased after 48 h of treatment. Numerous previous studies documented that several signaling pathways regulate MITF expression. MITF is phosphorylated by ERK and S6 [26,47]. Activation of ERK induces phosphorylation of MITF at Ser73, which leads to ubiquitination and degradation of the transcription factor [48]. In accordance with these previous reports, in treated vitiligo melanocytes, we found that phosphorylation of ERK induces MITF activation at the protein level. The dysfunction of intracellular metabolism in vitiligo may perturb and influence immune response. Vitiligo melanocytes release multiple cytokines and danger-associated molecular patterns, such as Hsp70 and HMGB1, in response to environmental signals relevant to the initiation of autoimmunity [5,49]. The relationships between the inflammatory milieu and metabolic impairment have always represented a key to understanding the complicated pathogenesis of vitiligo [28]. Furthermore, the stimulation of the immunological process may converge in terminal autoimmunity-mediated melanocyte loss. The amount of growing appreciation of similarities between vitiligo (autoimmunity) and melanoma (tumor immunity), has indicated the PD-1/PD-L1 axis as a promising target for future immunotherapies in human vitiligo [50]. A very novel observation of our current work is the capability of Pioglitazone to modulate several actors of the autoimmunity process. For the first time, here we reported a significant downregulation of PDL-1 by a PPARγ activator. Accordingly, PGZ reduces the expression of chemokine receptor CXCR3B, decreases the release of several DAMPs as well as, HMGB1 and Hsp70, and significantly lowers the gene expression levels of MICA and MICB. During the last years, our group reported that metabolic abnormalities are common to the entire vitiligo skin [7,8]. Starting from these considerations, we assayed that, in fibroblasts isolated from vitiligo patients, PGZ affected the release of growth factors and improved the production of extracellular components, such as Fibronectin. Furthermore, similar to precedent studies, we demonstrated that PPARγ agonist enhanced glycolysis and reverted the myofibroblast phenotype, characteristic of vitiligo dermal components. Moreover, PGZ treatment upregulated the gene and protein expression of CD36, a mediator of fatty acid uptake in both melanocytes and fibroblasts. This response can be explained by the fact that PGZ induces the accumulation of free fatty acids, as reported by Baranowski et al. [51]. The recent evaluation of the metabolic comorbidities in vitiligo patients [10], together with the known presence of pro-inflammatory cytokines linked to insulin resistance [52], defends the hypothesis of considering metabolic derangements as a new target to improve the quality of life of vitiligo patients. Unfortunately, vitiligo remains a difficult disease to treat. Taking into account all the different pathogenetic factors occurring in vitiligo onset, a therapeutical approach with a mechanism addressing the entire microenvironment seems to be needed. 

## 5. Conclusions

The encouraging results discussed above, including the tendency to restore the deranged metabolic profile, the decrease in senescence-related markers, and the amelioration trend of the inflammatory process, lead us to think of the possible use of a PPARγ agonist in the treatment of vitiligo. With in-depth research on targeted intrinsic metabolic abnormalities, we believe that the present research could represent an alternative therapeutic option to existing vitiligo treatments, generally providing only modest efficacy. Future studies will clarify our experimental furtherance in clinical practices.

## Figures and Tables

**Figure 1 cells-11-03583-f001:**
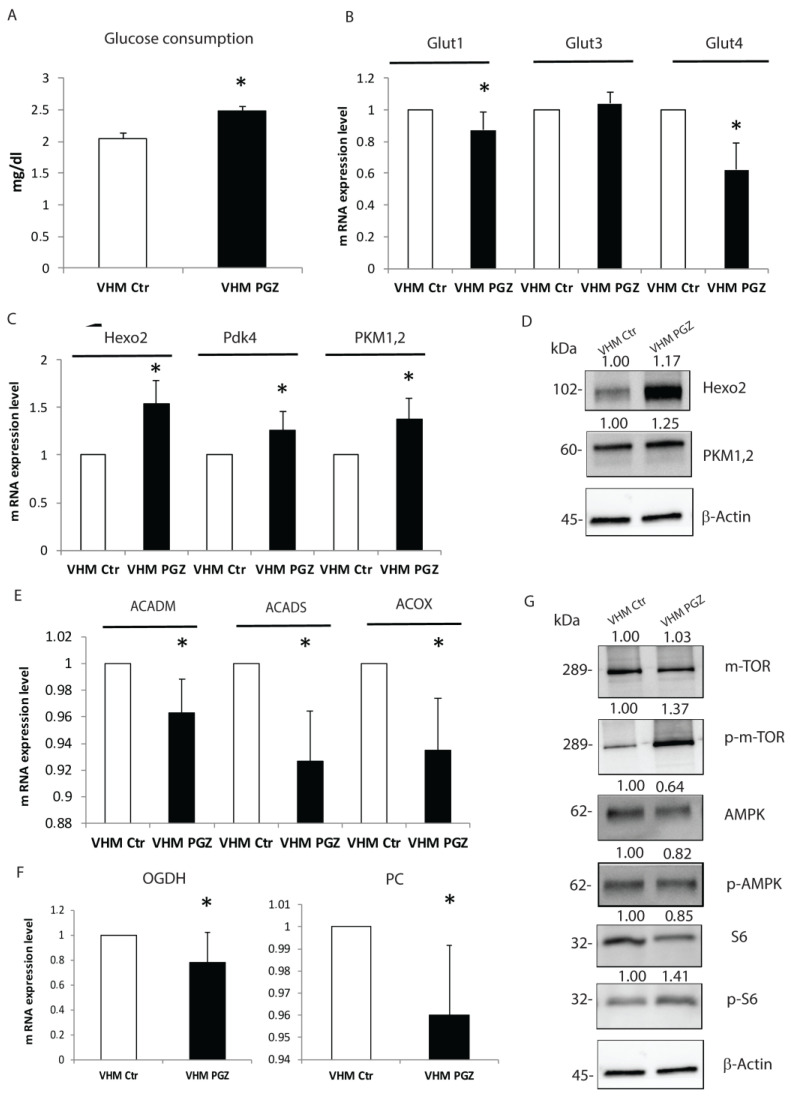
Pioglitazone modulates glucose metabolism in vitiligo melanocytes. (**A**) Two days of PGZ significantly increased glucose content in the vitiligo melanocytes medium. (**B**) PGZ reduced the gene expression of glucose transporters Glut1 and Glut4, but not Glut 3. (**C**,**D**) The transcriptional and protein levels of several anaerobic glycolytic enzymes were significantly induced by PGZ. mRNA levels of enzymes involved in β-oxidation (**E**) and Krebs cycle (**F**) were lowered by PGZ. (**G**) The representative Western blot of mTOR, pmTOR, AMPK, pAMPK, and S6, pS6, normalized to β-Actin, in vitiligo melanocytes following 2 days of treatment. The results are the mean ± SD of at least 3 independent experiments, * *p* ≤ 0.05.

**Figure 2 cells-11-03583-f002:**
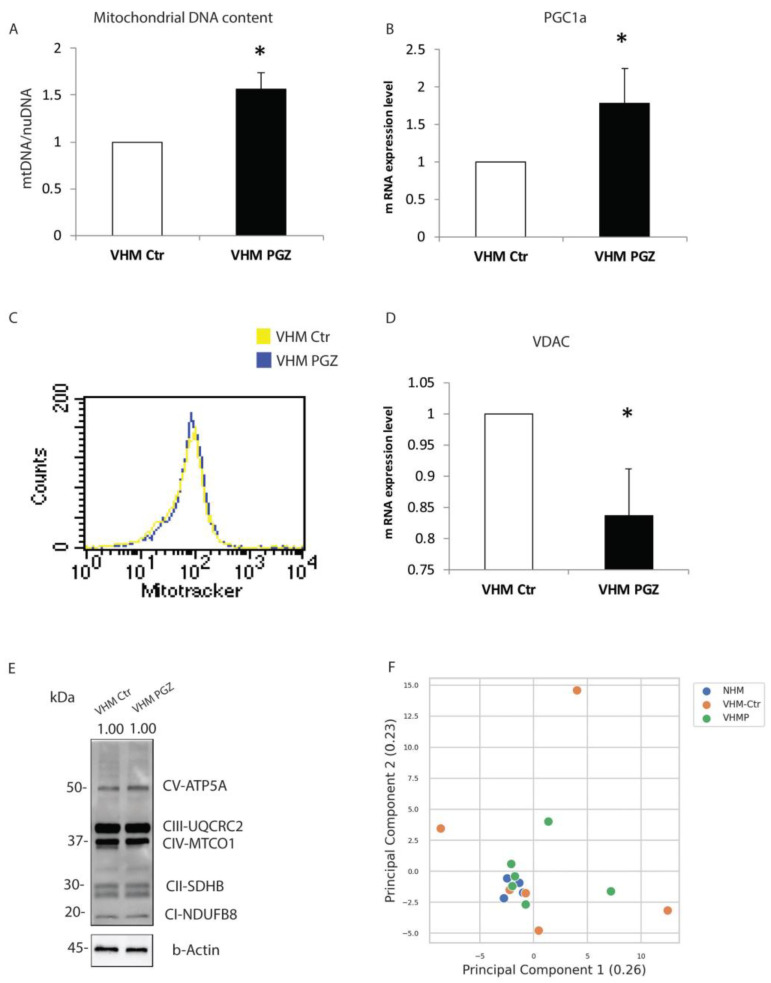
Pioglitazone increases the mitochondrial DNA integrity and copy number in vitiligo melanocytes. (**A**) Treatment with PGZ at 2 µM for 10 days significantly increased the mtDNA copy number. Nuclear DNA was used as an internal reference for data normalization. (**B**) RT-PCR underlined an upregulation of the PGC1α gene expression level. (**C**) The cytomic approach did not demonstrate any mitochondrial mass modification after PGZ treatment. (**D**) mRNA of VDAC1 was reduced by PGZ. (**E**) The representative immuno-blot did not show any modification of total OXPHOS complexes. (**F**) Principal component analysis (PCA) was generated by comparing 6 primary treated vitiligo melanocytes with the same number of primary untreated vitiligo and primary normal melanocytes. The data are mean ± SD of at least 3 independent experiments, * *p* ≤ 0.05.

**Figure 3 cells-11-03583-f003:**
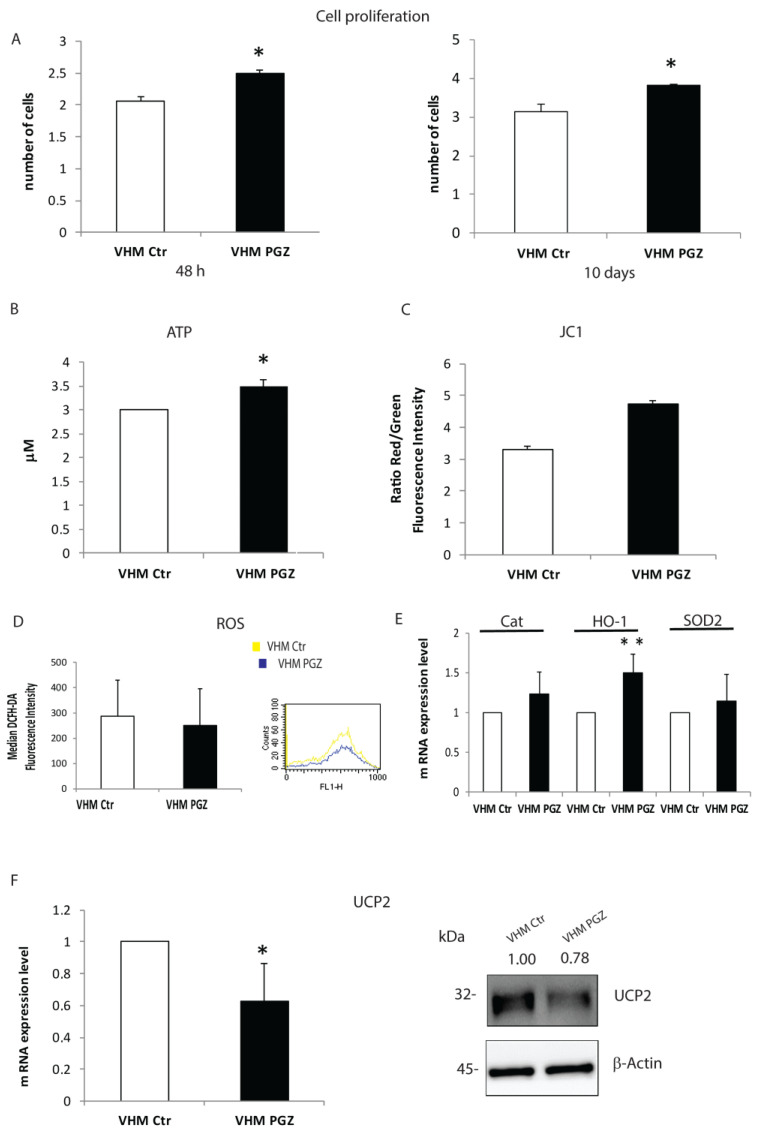
The impact of Pioglitazone on vitiligo melanocytes energetic status. (**A**) PGZ improved vitiligo melanocytes proliferation rate, evaluated by automatic cell counter, after 2 and 10 days of treatment. (**B**) Mitochondria-targeted luciferase assay revealed an increased ATP production in response to 10 days of Pioglitazone. (**C**) Flow cytometric analysis was used to quantify the improvement of the mitochondrial transmembrane potential of vitiligo cells. (**D**) The cytofluorimetric analysis highlighted a ROS reduction in treated vitiligo melanocytes. (**E**) Gene expression analysis of Cat, HO-1 and SOD2 was performed after 10 days of PGZ treatment. β-Actin expression was used to normalize the cDNA concentration. (**F**) RT-PCR analysis showed a significant down-modulation of UCP2, confirmed by Western blot analysis. The values are the mean ± SD of at least 3 independent experiments, * *p* ≤ 0.05 and ** *p* ≤ 0.01.

**Figure 4 cells-11-03583-f004:**
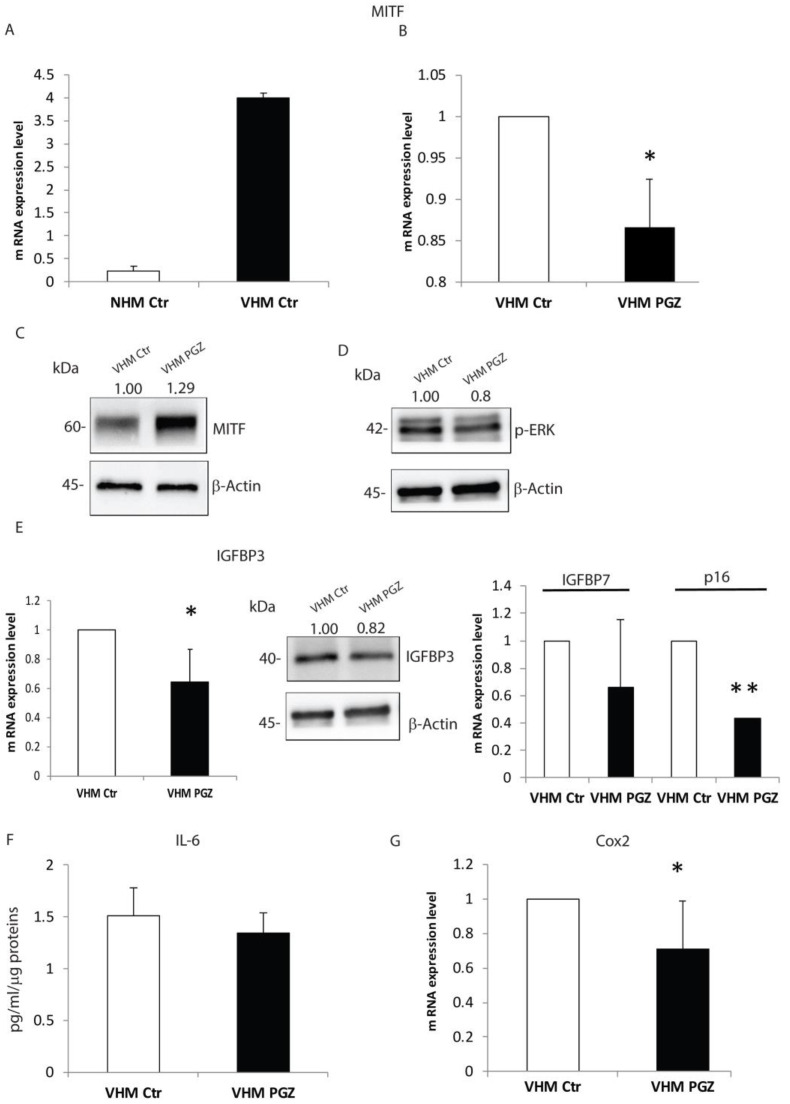
Pioglitazone modifies the expression of MITF and SASPs in vitiligo melanocytes. (**A**) Semi-quantitative real-time PCR was used to measure MITF mRNA expression in NHM and VHM. VHM showed a higher MITF expression level indicating a high grade of differentiation, which was reverted by PGZ treatment at mRNA (**B**), but not at protein level (**C**). (**D**) Representative Western blot of pErk. β-Actin was used as a loading control. (**E**) PGZ modulated the expression of senescence-associated markers, such as IGFBP3, both mRNA and protein levels, and IGFBP7 and p16 at mRNA level. (**F**) In vitiligo melanocytes, PGZ induced a reduction in IL-6 expression quantified by ELISA assay. (**G**) The relative Cox2 mRNA expression was evaluated after 6 h of treatment with PGZ (2 μM) and measured by qRT-PCR upon normalization to a reference gene (β-Actin), * *p* ≤ 0.05 and ** *p* ≤ 0.01.

**Figure 5 cells-11-03583-f005:**
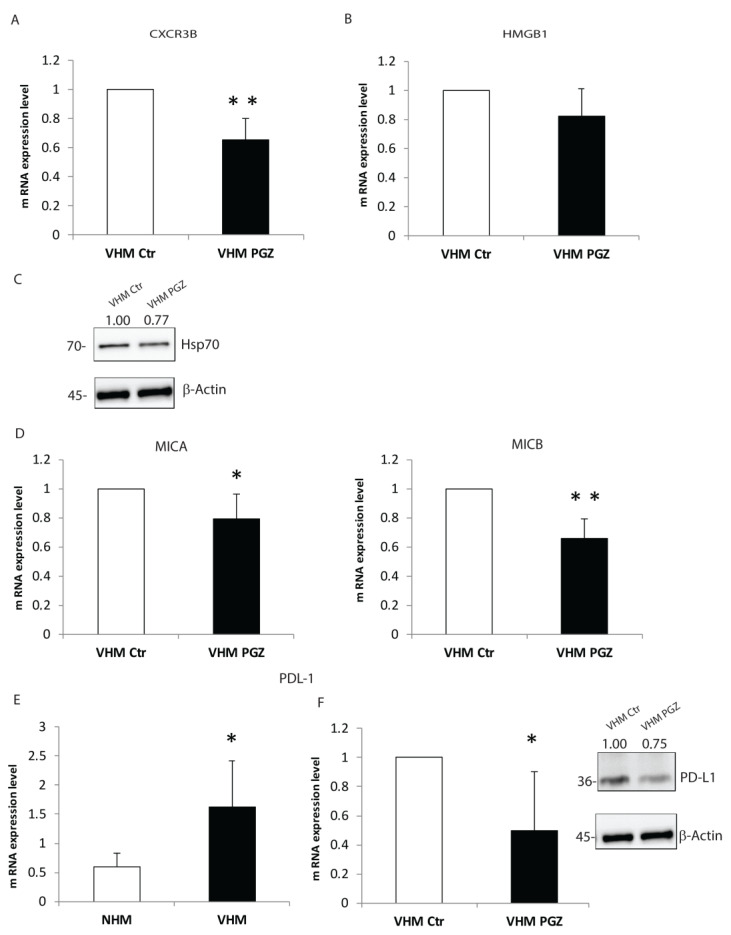
The mediation of pro-inflammatory agents by PGZ in vitiligo melanocytes. The mRNA level of chemokine receptor CXCR3B (**A**) and the inflammatory mediator HMGB1 (**B**) were investigated by RT-PCR after 6 h of treatment. (**C**) The representative protein appraisal of Hsp70. (**D**) mRNA evaluation of MICA and MICB, significantly downregulated by PGZ. (**E**) PDL-1 gene expression comparison between normal and vitiligo melanocytes. (**F**) mRNA downregulation of PDL-1 in vitiligo melanocytes by PGZ treatment, confirmed by immunoblotting assay. The data are mean ± SD of at least 3 independent experiments, * *p* ≤ 0.05 and ** *p* ≤ 0.01.

**Figure 6 cells-11-03583-f006:**
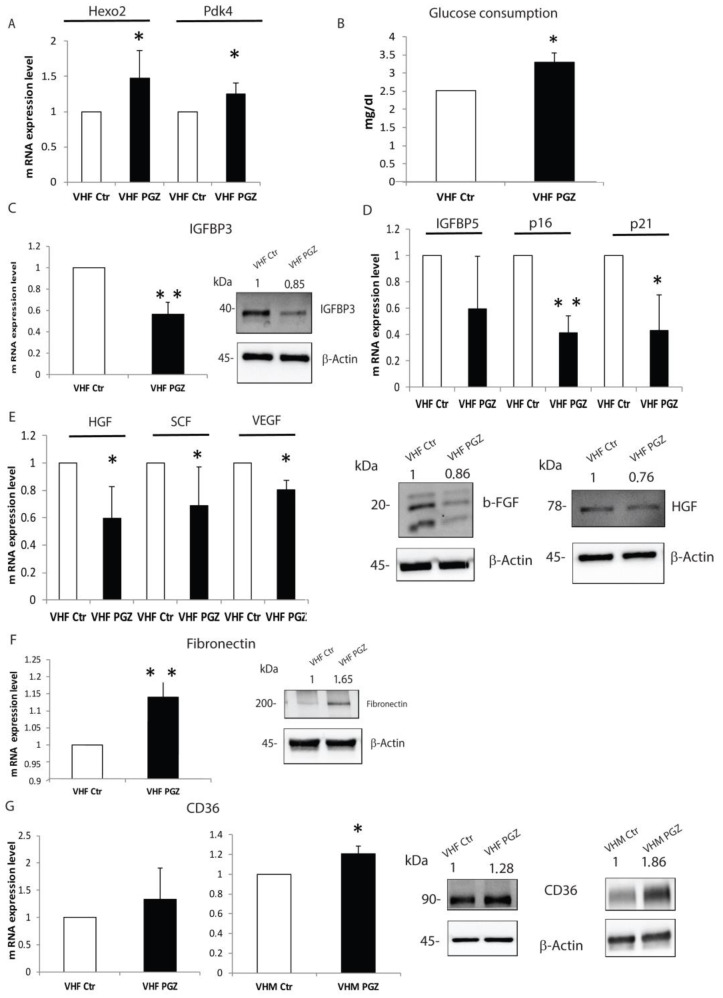
PGZ treatment is also effective on the dermal component. (**A**) The evaluation of gene expression of several anaerobic glycolytic enzymes induced by PGZ. (**B**) Significant glucose increase content in the medium of vitiligo fibroblasts. (**C**) IGFBP3 was downregulated in vitiligo fibroblasts by PGZ, both mRNA and protein levels. (**D**) Gene expression downregulation of senescence markers after 6 h of PGZ treatment. (**E**) The growth factors gene expression was analyzed by RT-PCR. The results were shown as a fold increase above the control. b-FGF and HGF were also assessed by Western blot. (**F**) The overexpression of Fibronectin was detected by RT-PCR and Western blot in vitiligo fibroblasts, respectively, after 6 and 48 h of PGZ treatment. (**G**) CD36 gene expression in vitiligo fibroblasts and melanocytes after PGZ treatment. The representative Western blot of CD36 total protein in vitiligo cell lines. All data are expressed as means ± SD of at least 3 independent experiments, * *p* ≤ 0.05 and ** *p* ≤ 0.01.

## Data Availability

Not applicable.

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
