# Peer review of "A Possible Modulator of Vitiligo Metabolic Impairment: Rethinking a PPARγ Agonist"

_cells, 2022, doi:10.3390/cells11223583_

Round 1

Reviewer 1 Report

Papaccio et al. presented an interesting and novel research of pioglitazone, a PPAR agonist on its modulation of vitiligo metabolic impairment by in vitro studies on melanocytes and fibroblasts from vitiligo patients. The results are abundant and inspiring. And here are several suggestions that might help further clarify the main idea and achieve better preciseness.

1.      In the study, the PGZ were treated from 6h to 10 days, and the primary melanocytes were performed between 2 and 7 culture passages. Factors associated with glucose metabolism were mostly tested after 6h-48h treatment, while the mitochondria-associated results, including mtDNA copys, PGC and VDAC1 mRNA level were all measured after 10 days treatment. Why is the difference? How did the author further explain the different impacts of acute and prolonged treatment of PGZ on melanocytes? Besides, A cell proliferation assay of melanocytes treated by PGZ for 10 days might be needed, to assure the viability of the relatively long-time treatment of primary melanocytes. (The current version has included the cell viability results for 48h treatment only.)

2.      The subtitles of the results part might need further consideration and keep coherence with the figure legends. For example, line 376, the following part is mainly about autoimmunity rather than “the release of SASPs and inflammatory mediators in vitiligo melanocytes”. Besides, it seems not solid enough to draw the conclusion of PGZ’s modulation on SASPs simply by detection of IGFBP3 (a metabolic-related molecular), IL-6 and COX-2 (including in senescence but also inflammation).

3.      There is a paradoxical description in line 236-237,” Pioglitazone did not improve the rate of glucose transporters Glut1, Glut3, and Glut 4 as confirmed at both mRNA and protein levels, respectively after 6 and 48 hours (Fig.1B). “and the figure legend “PGZ reduced the gene expression of glucose transporters Glut1 and Glut4, but not Glut 3. “ and the protein level of Glut is not shown in the Figure 1B.

4.      As for figures of glucose consumption (Fig 1A and 6A), the figure legend and the graph didn’t show the same trend, possibly due to the detection is based on the residual glucose concentration (the lower left, the higher consumption). Still, it might be a little confused for readers, especially under the title of glucose consumption.

5.      The conclusion part of the article might be better to summarize the results of the current research. The outlook and future plans might suit better in discussion part.

Author Response

Reviewer 1

Papaccio et al. presented an interesting and novel research of pioglitazone, a PPAR agonist on its modulation of vitiligo metabolic impairment by in vitro studies on melanocytes and fibroblasts from vitiligo patients. The results are abundant and inspiring. And here are several suggestions that might help further clarify the main idea and achieve better preciseness.

  1. In the study, the PGZ were treated from 6h to 10 days, and the primary melanocytes were performed between 2 and 7 culture passages. Factors associated with glucose metabolism were mostly tested after 6h-48h treatment, while the mitochondria-associated results, including mtDNA copys, PGC1αand VDAC1 mRNA level were all measured after 10 days treatment. Why is the difference? How did the author further explain the different impacts of acute and prolonged treatment of PGZ on melanocytes? Besides, A cell proliferation assay of melanocytes treated by PGZ for 10 days might be needed, to assure the viability of the relatively long-time treatment of primary melanocytes. (The current version has included the cell viability results for 48h treatment only.)

      A: Thank you for the comment. Our aim was to evaluate the long-term effects of the treatment. However, all genes were detected after 6 hours of treatment, including PGC1a and VDAC. The other parameters, such as mitochondrial DNA, ATP production and mitochondrial membrane potential require long-term treatment to modify their status. As suggested, we added in Figure 3 a cell proliferation assay of melanocytes treated by PGZ for 10 days.

  1. The subtitles of the results part might need further consideration and keep coherence with the figure legends. For example, line 376, the following part is mainly about autoimmunity rather than “the release of SASPs and inflammatory mediators in vitiligo melanocytes”. Besides, it seems not solid enough to draw the conclusion of PGZ’s modulation on SASPs simply by detection of IGFBP3 (a metabolic-related molecular), IL-6 and COX-2 (including in senescence but also inflammation).

A: According to the Reviewer, we modified the subtitle and improved the detection of senescence-related markers in vitiligo melanocytes and fibroblasts, evaluating the expression of IGFBP5, IGFBP7, p21 and p16.

  1. There is a paradoxical description in line 236-237,”Pioglitazone did not improve the rate of glucose transporters Glut1, Glut3, and Glut 4 as confirmed at both mRNA and protein levels, respectively after 6 and 48 hours (Fig.1B). “and the figure legend “PGZ reduced the gene expression of glucose transporters Glut1 and Glut4, but not Glut 3. and the protein level of Glut is not shown in the Figure 1B.

      A: We apologise for the mistake, and we have corrected the sentences.

  1. As for figures of glucose consumption (Fig 1A and 6A), the figure legend and the graph didn’t show the same trend, possibly due to the detection is based on the residual glucose concentration (the lower left, the higher consumption). Still, it might be a little confused for readers, especially under the title of glucose consumption.

      A: We tried to clarify the results in the test and in the figures.

  1. The conclusion part of the article might be better to summarize the results of the current research. The outlook and future plans might suit better in discussion part.

     A: As suggested, we summarized the results in the conclusion part and moved the outlook and future plans in the end of the discussion part.

Reviewer 2 Report

Decision

Major Revision

Comments

It has been reported the conducted a study to evaluate the effect of Pioglitazone, PPARγ agonist, on cells from pigmented vitiligo skin, it belongs to new use of old medicine. Although this observation is potential interesting, there are some serious issues in the manuscript related to the experimental design, writing, interpretation, and analyses should be addressed before publication. Therefore, the work is preliminary and the data presented in this manuscript don’t provide strong evidence for their conclusions at this present form.

Points of concern that need to be addressed by the authors are as follows:

1. Results section of line 224, the cited paper (ref [6] Dell'Anna ML. et al., 2017) analyze the mitochondrial failure rescue by cardiolipin manipulation may be a new intriguing target in treatment development. But the relationship between vitiligo and anaerobic glycolytic enzymes which the author claimed should provide more evidence to verification in this study. Why this happening, and what is the explanation of the author against this result. This phenomenon should be properly discussed in the manuscript.

2. It is very important for choose model (cell or animal) used for explore drug function and effective mechanism, so it should take this seriously. The ratio of melanocytes to keratinocyte in basal layer is vary 1: 10 to 1: 5 in different dissect location of the skin; the density of melanocytes is highest in the epidermal region of facial and reproductive organ. Vitiligo lesions lack melanocytes and benefit from treatments that reduce melanocyte death and/or promote melanocyte regrowth. Impaired melanogenesis should not be considered a primary deficiency of vitiligo lesions, as the introduction and discussion claim. Therefore: 1) The author obtained three human primary cell lines: normal human primary epidermal melanocytes (NHM), primary epidermal melanocytes (VHM) and fibroblasts (VHF). Dose establish immortalized experimental conducted with these isolated cells? Dose these isolated cell lines authenticated by the third detection agency? If not, it should ensure stability of cell passage, so the subsequent experimental results were influenced. 2) Why do not purchase proposed human cell line PIG1and PIG3V from specialist agencies? Because the samples from clinical are do not used for drug investigation and mechanism exploration.

3. Theme of the manuscript put forword to evaluate role of PPARγ pathway with PGZ, then, the experimental design and Discussion should focus on the mechanism of PPARγ pathway and action of PGZ. Complementary functional recovery experiment is recommended. For example, after PPARγ is silenced / overexpressed in melanocytes, ROS level is suggested overexpressed or silenced, and then pathway related genes and protein should be detected.

4. It need for focus on how to restore viable melanocytes in affected areas or efficacy of melanosome transport to keratinocytes in the skin. The impact of PGZ on melanogenesis is more worthy for evaluation. The author did not show evidence for such as tyrosinase activity and melanin content. Is the PGZ increase the melanin synthesis in melanocytes? Did PGZ increase the cell proliferation, gene and protein expression dose dependently? If not, the conclusion should be revised carefully.

5. Here are abundant grammatical errors throughout the manuscript, which hinder the readers to understand authors' message in this current form. Extensive and thorough English editing is indispensable. Such as inaccurate terms “vitiligo skin”, “vitiligo cells” and “vitiligo melanocytes”, dose the author want to express the human primary epidermal melanocytes isolated from lesional skin? And “PPARγ” and “PPAR-γ”. Please modify.

6. The abbreviations should be used with full names only for the first time, after they can be directly used in the text, e.g. “pioglitazone (PGZ)”, “Primary epidermal melanocytes (VHM)” should be changed in the following statement; please check the whole manuscript carefully.

Author Response

Reviewer 2

Major Revision

Comments

It has been reported the conducted a study to evaluate the effect of Pioglitazone, PPARγ agonist, on cells from pigmented vitiligo skin, it belongs to new use of old medicine. Although this observation is potential interesting, there are some serious issues in the manuscript related to the experimental design, writing, interpretation, and analyses should be addressed before publication. Therefore, the work is preliminary and the data presented in this manuscript don’t provide strong evidence for their conclusions at this present form.

Points of concern that need to be addressed by the authors are as follows:

  1. Results section of line 224, the cited paper (ref [6] Dell'Anna ML. et al., 2017) analyze the mitochondrial failure rescue by cardiolipin manipulation may be a new intriguing target in treatment development. But the relationship between vitiligo and anaerobic glycolytic enzymes which the author claimed should provide more evidence to verification in this study. Why this happening, and what is the explanation of the author against this result. This phenomenon should be properly discussed in the manuscript.

  1. It is very important for choose model (cell or animal) used for explore drug function and effective mechanism, so it should take this seriously. The ratio of melanocytes to keratinocyte in basal layer is vary 1: 10 to 1: 5 in different dissect location of the skin; the density of melanocytes is highest in the epidermal region of facial and reproductive organ. Vitiligo lesions lack melanocytes and benefit from treatments that reduce melanocyte death and/or promote melanocyte regrowth. Impaired melanogenesis should not be considered a primary deficiency of vitiligo lesions, as the introduction and discussion claim. Therefore: 1) The author obtained three human primary cell lines: normal human primary epidermal melanocytes (NHM), primary epidermal melanocytes (VHM) and fibroblasts (VHF). Dose establish immortalized experimental conducted with these isolated cells? Dose these isolated cell lines authenticated by the third detection agency? If not, it should ensure stability of cell passage, so the subsequent experimental results were influenced. 2) Why do not purchase proposed human cell line PIG1and PIG3V from specialist agencies? Because the samples from clinical are do not used for drug investigation and mechanism exploration.

A: Thank you for the comment. Usually, we use primary cell lines obtained from healthy and/or vitiligo patients. As for the vitiligo cells, biopsies are always carried out starting from the same non-photo-exposed and pigmented area, making the samples uniform. The experiments are carried out between the third and seventh culture passage. The use of primary cell lines could reproduce in vitro the subject variability, which remains an important characteristic of vitiligo. Moreover, immortalized cell lines are not a useful model to study senescent phenomena.

  1. Theme of the manuscript put forward to evaluate role of PPARγ pathway with PGZ, then, the experimental design and Discussion should focus on the mechanism of PPARγ pathway and action of PGZ. Complementary functional recovery experiment is recommended. For example, after PPARγ is silenced / overexpressed in melanocytes, ROS level is suggested overexpressed or silenced, and then pathway related genes and protein should be detected.

A: Thank you for the comment. The suggestion is very interesting.  PPARg silencing is a transient transfection, which would not allow us to make a meaningful assessment of ROS. However, in agreement with the reduction of ROS levels, we added the evaluation of gene expression of several antioxidant enzymes after 10 days of PGZ treatment.

  1. It need for focus on how to restore viable melanocytes in affected areas or efficacy of melanosome transport to keratinocytes in the skin. The impact of PGZ on melanogenesis is more worthy for evaluation. The author did not show evidence for such as tyrosinase activity and melanin content. Is the PGZ increase the melanin synthesis in melanocytes? Did PGZ increase the cell proliferation, gene and protein expression dose dependently? If not, the conclusion should be revised carefully.

A: We thank the Reviewer for this suggestion. However, the aim of our study is to restore the metabolic impairment of cells in the pigmented skin of vitiligo patients. Our research activity has been particularly focused on the elucidation of nonimmunological factors in the pathogenesis of vitiligo, trying to identify inherited defects in cells of the normal-appearing skin of vitiligo patients which could be the basis for the increased sensitivity of melanocytes to stressful events. Our hypothesis is that metabolic impairment concurs in autoimmunity response because can be the basis for the release of HSP70 and other DAMPS which activate innate and then adaptative immune response. The present research work, therefore, did not have as its ultimate goal the repigmentation but the restoring of the altered metabolism that could influence different pathways, causing the manifestation of the disease. The melanogenesis process has not been investigated, since in vitiligo the white patches appearance is the consequence of melanocyte loss rather than alteration of melanogenesis.

  1. Here are abundant grammatical errors throughout the manuscript, which hinder the readers to understand authors' message in this current form. Extensive and thorough English editing is indispensable. Such as inaccurate terms “vitiligo skin”, “vitiligo cells” and “vitiligo melanocytes”, dose the author want to express the human primary epidermal melanocytes isolated from lesional skin? And “PPARγ” and “PPAR-γ”. Please modify.

A: Thank you for the comment, we corrected the grammatical errors. The manuscript has been revised by an English language expert. With the terms “vitiligo skin”, “vitiligo cells” and “vitiligo melanocytes”, we want express vitiligo cells, melanocytes, or fibroblasts isolated from the pigmented area of vitiligo patients.

  1. The abbreviations should be used with full names only for the first time, after they can be directly used in the text, e.g. “pioglitazone (PGZ)”, “Primary epidermal melanocytes (VHM)” should be changed in the following statement; please check the whole manuscript carefully.

A: We thank the Reviewer for this observation, we did not notice it. We checked the abbreviation in the whole manuscript and modified the text accordingly.

Reviewer 3 Report

In this manuscript, Papaccio and colleagues investigate whether the Insulin sensitizer and PPAR gamma agonistic drug Pioglitazone would affect cells from vitiligo patients. These cells display, in vivo and in culture, metabolic aberration and markers of cellular stress or an early senescent phenotype which is regarded causative or supporting of the downstream disruption of pigment production and inflammation. Due to the not fully resolved pathogenesis of vitiligo, there is still the need for safe and specific treatment options that are not covered by the existing treatment strategies. Especially the co-occurence of vitiligo with markers of metabolic syndrome recognized by this working group makes the anti diabetic drug Pioglitazone an interesting candidate and at the same time research on the metabolic component of vitiligo may give novel insights into the diseases etiology.

This manuscript and it’s implication have major merit, novelty and will have important impact, but the presentation and interpretation of the results needs clarification, re-phrasing and caution, and a general utilization of better quality figures and fonts.

Specific points, major and minor

The presentation of the glucose consumption Fig 1 a  and the corresponding legend and text should be adapted for a general readership, explaining (i guess) that the increased glucose measurement in the media is an indicator for decreased consumption. Also the interpretation of other results presented in figure 1 needs to be addressed. Example “To further corroborate our analysis, we 246 conducted a measure of b-oxidation key enzymes, such as Acyl-CoA Dehydrogenase me- 247 dium-chain (ACADM), Acyl-CoA Dehydrogenase (ACADS), Peroxisomal Acyl-CoA Ox- 248 idase (ACOX). We found that Pioglitazone didn’t modulate this class of genes (Fig. 1F), 249 confirming PGZ selective activity on pathways implicated in ATP production”  The corresponding figure does however show a significant decrease in the mRNA expression of these enzymes on mRNA level, comparable to the increase of the Hexokinase and pyruvate kinase shown before. (it also should read Fig 1 E , and the genes measured in 1F are not described in unabbreviated form)

ad Figure 2 minor: Is the copy number assesed by absolute quantification? it rathers seems that this represents a relative increase in mtDNA normalized to nuclear DNA, then it should also be stated so.

Figure 2F – The dataset and method used for the PCA analysis should be specified in more detail. It was performed only on those genes shown individually in the study?  Furthermore, for readers with expertise in senescence research, the far distributioin of the PCA of the vitiligo samples is well conceivable, but for the general readership the novel concepts of heterogeneity in senescent(type) vitiligo cells and specimens should be explained and cited.  

Minor: the fonts need to be increased in 2F

Also Figure 3 needs to be checked for consistency

while the figure legend reads

“(D) Cytofluorimetric analysis highlighted a ROS 315 reduction in treated vitiligo melanocytes. (E) RT-PCR analysis showed a significant upregulation of 316 UCP2, confirmed by western blot analysis.”

This is in contrast with the presentation of the bars in Figure D which show alsmost identical values of DCFH-DA intensity. And in panel E, the RT PCR analysis shows a decrease, not an increase of UCP2

Minor : In Figure 4 the panel numbering/naming is not correct.

“(D) The figure indicates that 347 Pioglitazone modulates the expression of a senescence-associated marker, such as insulin-like 348 growth factor-binding protein 3 (IGFBP3), in vitiligo melanocytes both mRNA and protein levels. 349 (E) In vitiligo melanocytes, PGZ induces a reduction in IL-6 expression quantified by ELISA assay 350 (F) Relative Cox2 mRNA expression was evaluated after 6 hours of treatment with PGZ (2 μM) and 351 measured by qRT-PCR upon normalization to a reference gene (b-Actin)”

(G is missing, others are wrongly named)

Author Response

Reviewer 3

In this manuscript, Papaccio and colleagues investigate whether the Insulin sensitizer and PPAR gamma agonistic drug Pioglitazone would affect cells from vitiligo patients. These cells display, in vivo and in culture, metabolic aberration and markers of cellular stress or an early senescent phenotype which is regarded causative or supporting of the downstream disruption of pigment production and inflammation. Due to the not fully resolved pathogenesis of vitiligo, there is still the need for safe and specific treatment options that are not covered by the existing treatment strategies. Especially the co-occurence of vitiligo with markers of metabolic syndrome recognized by this working group makes the anti diabetic drug Pioglitazone an interesting candidate and at the same time research on the metabolic component of vitiligo may give novel insights into the diseases etiology.

This manuscript and it’s implication have major merit, novelty and will have important impact, but the presentation and interpretation of the results needs clarification, re-phrasing and caution, and a general utilization of better quality figures and fonts.

Specific points, major and minor

The presentation of the glucose consumption Fig 1 a  and the corresponding legend and text should be adapted for a general readership, explaining (i guess) that the increased glucose measurement in the media is an indicator for decreased consumption. Also the interpretation of other results presented in figure 1 needs to be addressed. Example “To further corroborate our analysis, we 246 conducted a measure of b-oxidation key enzymes, such as Acyl-CoA Dehydrogenase me- 247 dium-chain (ACADM), Acyl-CoA Dehydrogenase (ACADS), Peroxisomal Acyl-CoA Ox- 248 idase (ACOX). We found that Pioglitazone didn’t modulate this class of genes (Fig. 1F), 249 confirming PGZ selective activity on pathways implicated in ATP production”  The corresponding figure does however show a significant decrease in the mRNA expression of these enzymes on mRNA level, comparable to the increase of the Hexokinase and pyruvate kinase shown before. (it also should read Fig 1 E , and the genes measured in 1F are not described in unabbreviated form)

A: According to the Reviewer, we corrected the sentences and the corresponding figure legend.

ad Figure 2 minor: Is the copy number assesed by absolute quantification? it rathers seems that this represents a relative increase in mtDNA normalized to nuclear DNA, then it should also be stated so.

A: Thank you for the comment. The mitochondrial DNA content is normalized to nuclear DNA. As suggested, we stated it.

Figure 2F – The dataset and method used for the PCA analysis should be specified in more detail. It was performed only on those genes shown individually in the study?  Furthermore, for readers with expertise in senescence research, the far distribution of the PCA of the vitiligo samples is well conceivable, but for the general readership the novel concepts of heterogeneity in senescent(type) vitiligo cells and specimens should be explained and cited.  

A: We thank the Reviewer for this suggestion. We specified in more detail the dataset and method used for the PCA analysis in the Materials and Methods section. Regarding the far distribution of vitiligo samples in the PCA analysis, we added a sentence and a citation, explaining the heterogeneity of vitiligo cells.

Minor: the fonts need to be increased in 2F

 A: As suggested, we increased the front of 2F PCA image

Also Figure 3 needs to be checked for consistency

while the figure legend reads

“(D) Cytofluorimetric analysis highlighted a ROS 315 reduction in treated vitiligo melanocytes. (E) RT-PCR analysis showed a significant upregulation of 316 UCP2, confirmed by western blot analysis.”

This is in contrast with the presentation of the bars in Figure D which show alsmost identical values of DCFH-DA intensity. And in panel E, the RT PCR analysis shows a decrease, not an increase of UCP2

A: Thank you for the comment. We checked the results and corrected the wrong representative images and histogram.

Minor : In Figure 4 the panel numbering/naming is not correct.

“(D) The figure indicates that 347 Pioglitazone modulates the expression of a senescence-associated marker, such as insulin-like 348 growth factor-binding protein 3 (IGFBP3), in vitiligo melanocytes both mRNA and protein levels. 349 (E) In vitiligo melanocytes, PGZ induces a reduction in IL-6 expression quantified by ELISA assay 350 (F) Relative Cox2 mRNA expression was evaluated after 6 hours of treatment with PGZ (2 μM) and 351 measured by qRT-PCR upon normalization to a reference gene (b-Actin)”

(G is missing, others are wrongly named)

A: We thank the reviewer and we corrected the naming and the numbering in the figures and in the text, and modified the figure legend.

Round 2

Reviewer 2 Report

Decision

Accept

Comments

The author gives serious responses to all the reviewer's questions, in addition, the manuscript revised sufficiently.

I think the opportunity for Accept may be more reasonable.

Reviewer 3 Report

All concerns by this reviewer have been adressed